# Isotropic and Anisotropic Scaffolds for Tissue Engineering: Collagen, Conventional, and Textile Fabrication Technologies and Properties

**DOI:** 10.3390/ijms22179561

**Published:** 2021-09-03

**Authors:** Robert Tonndorf, Dilbar Aibibu, Chokri Cherif

**Affiliations:** Institute of Textile Machinery and High Performance Material Technology, Technische Universität Dresden, 01069 Dresden, Germany; dilbar.aibibu@tu-dresden.de (D.A.); chokri.cherif@tu-dresden.de (C.C.)

**Keywords:** scaffold, biomaterials, biopolymers, collagen, chitosan, pores, fibers, anisotropic, textile, spinning

## Abstract

In this review article, tissue engineering and regenerative medicine are briefly explained and the importance of scaffolds is highlighted. Furthermore, the requirements of scaffolds and how they can be fulfilled by using specific biomaterials and fabrication methods are presented. Detailed insight is given into the two biopolymers chitosan and collagen. The fabrication methods are divided into two categories: isotropic and anisotropic scaffold fabrication methods. Processable biomaterials and achievable pore sizes are assigned to each method. In addition, fiber spinning methods and textile fabrication methods used to produce anisotropic scaffolds are described in detail and the advantages of anisotropic scaffolds for tissue engineering and regenerative medicine are highlighted.

## 1. Introduction

The two terms “regenerative medicine” and “tissue engineering” are used synonymously as well as differently in the current literature. It is difficult to distinguish between both terms, as their descriptions have a great deal of intersection. The ideas and concepts of “regenerative medicine” and “tissue engineering” have been used since the beginning of the 20th century: Alexis Carrel was developing techniques to cultivate cells in vitro and was proposing with Charles Lindbergh to grow organs as early as the 1930s [1]. These concepts were focused exclusively on cells until the 1970s. By discovering the importance of the extracellular matrix in the 1970s, the concept of “tissue engineering” was established in the following decades. The term “regenerative medicine” has gained in importance with the developments in stem cell research since the 2000s [2]. If a differentiated view of the two terms is indispensable, “tissue engineering” may be assigned to the engineering context and “regenerative medicine” to the medical or biological context. Likewise, both terms can be summarized as “tissue engineering and regenerative medicine” (TERM).

## 2. Tissue Engineering and Regenerative Medicine (TERM)

### 2.1. Fundamentals of Tissue Engineering and Regenerative Medicine

The TERM approach is based on the interaction of up to three components (triad). New tissue forms by cultivating cells on cell carriers or scaffolds under the influence of chemical or physical signals (Figure 1).

The cells used in the TERM are mostly stem cells. These cells can self-renew in terms of proliferation and generate somatic cells, which can proliferate only to a very limited extent. The relevant stem cells for TERM are mesenchymal stem cells (MSCs) [3], embryonic stem cells (ESCs) [4], and induced pluripotent stem cells (iPSCs) [5,6].

Chemical signals such as growth factors influence the proliferation and differentiation of cells. However, as the bioactivity of growth factors has a short half-life in physiological environments, delivery systems are developed for on-demand growth factor delivery [7,8,9]. Physical signals are equally a focus of research as moderate mechanical stimulation ensures maintenance of the natural phenotype of tissues [10]. Mechanical stimulation methods are therefore used to generate a specific tissue phenotype in TERM such as cartilage [11], tendon [12], or cardiac muscle tissue [13].

Scaffolds serve as cell carriers in the form of artificial extracellular matrices (ECMs). They determine the geometric template for the tissue to be generated by stabilizing cells and, if necessary, signal molecules in a defined manner in three-dimensional space. Scaffolds must have an open-pore structure to allow migration of cells and nutrients. Many scaffolds employ pore sizes between 100 and 400 µm, with pore sizes varying depending on the specific cell and tissue type [14]. Therefore, the fabrication method of scaffolds must enable adjustability of the pore size. In addition, scaffolds control the proliferation and differentiation potential of cells through their material-specific properties such as surface chemistry, roughness, structuring, and stiffness (mechanotransduction) [15,16,17]. After successful proliferation, differentiation, tissue formation, and optional vascularization, the scaffold material must degrade without leaving residues. Scaffold materials can be of either synthetic or natural origin, but they have to possess certain characteristics to be suitable for TERM. These include good biocompatibility, biodegradability, biomimetics (a structure and mechanical behavior similar to that of natural ECM), and sufficiently high mechanical stability under physiological conditions.

### 2.2. In Vivo, Ex Vivo, In Vitro, and In Situ

The successive steps of cell culture: seeding, proliferation, differentiation, and tissue formation can take place either inside or outside the body. In case of the in vitro strategy, cells are obtained from the patient (autologous transplantation) or a donor (allogeneic transplantation), which are then seeded on a scaffold. After proliferation and differentiation, new tissue is formed, which is implanted in the defect of the body [18]. For the in vivo strategy, cells are obtained analogous to the in vitro strategy. However, proliferation and differentiation are started after seeding and implantation. For this purpose, the scaffold seeded with cells is implanted either directly in the defect site or at another site in the body, with the body serving as a bioreactor in each case [19]. In the in situ strategy, on the other hand, an acellular scaffold (without cells) is implanted into the defect site. Subsequently, the new tissue is formed by regenerative processes of the body after cells from the adjacent tissue migrate into the scaffold, proliferate, and differentiate [20]. However, a clear classification of the different strategies is difficult, as no clear boundary is drawn in the literature, especially between the in vivo and in situ strategies.

In ex vivo cell culture, cells are cultivated outside a living organism under controlled conditions. The ex vivo cell culture technology is applied in the in vitro and in vivo strategy. Hence, a high availability of stem cells and logistic efforts are required. In addition, organ and tissue damage may occur in the donor when the cells are harvested. Additionally, phenotypic changes of the cells may occur. Last but not least, autocrine or paracrine signaling (hormone release from cells to themselves or to neighboring cells) is difficult to implement, especially with the in vitro strategy. The in vivo cell culture of the in situ strategy decreases these limitations and challenges [21,22,23].

Regulatory challenges also need to be considered. The European Regulation No. 1394/2007 describes the regulations for cell-containing medical devices (“advanced therapy medicinal products,” ATMPs) and became effective in 2008. The ATMPs include the three product classes: gene therapy, somatic cell therapy, and tissue engineered products. Thus, high requirements for approval are set for implants applied in the in vitro strategy and in vivo strategy. These regulations do not apply to cell-free scaffolds from the in situ strategy, which gives them an advantage with regard to approval.

In summary, due to the disadvantages resulting from the stem cell harvesting, ex vivo cell culture and regulatory challenges, cell-free scaffolds for the in situ strategy of TERM have increasingly become the focus of research and development.

### 2.3. Tissue Engineering Applications

The natural or physiological regeneration and repair of tissues is a complicated and continuous process in all living beings. In regeneration, the defective tissue is replaced by new tissue and cells, whereas in repair, normal or pathological scar tissue is generated. Complete regeneration occurs only in some animal species (e.g., salamanders, zebrafish, and reindeer). In human embryos, complete regeneration is found until the last trimester of pregnancy using endogenous stem cells. Which of the two processes dominates depends on different growth factors and cytokines [24,25,26].

The need for TERM originates from the inability of the human body to regenerate tissue. The most important tissues for TERM include cartilage, skin, bone, cornea, nerves, tendons, ligaments, cardiac tissues (e.g., arteries and heart valves), and teeth (Figure 2) [27]. Of these tissue types, cartilage tissue has a particularly poor physiological regenerative capacity. This is due to the avascular nature (the lack of blood vessels) of cartilage, the limited availability of chondrocytes (cartilage cells), and their limited proliferative potential [28].

## 3. Biomaterials

### 3.1. Overview

Scaffolds are made of biomaterials, being characterized by their biocompatibility (i.e., not causing any negative effects on the metabolism of living organisms when in direct contact with them). According to recent systematic reviews, biomaterials may be polymers, ceramics, metals, and their composites; they can be of synthetic or biological origin and with and without the ability to degrade [29,30,31,32]. However, non-degradable biomaterials are used for the fabrication of permanent implants (tissue substitutes), whereas biodegradable materials are used for scaffolds as their support for tissue formation is only temporary. Therefore, the biocompatibility of biomaterials for implants is essentially based on their inertness (i.e., little or no interaction between biomaterial and tissue is required). The biocompatibility of biomaterials for scaffolds is primarily based on providing an environment for cells similar to the natural ECM (biomimetics) and on their biodegradability. In addition to the biological requirements, these biomaterials must provide sufficient stability in aqueous media as they must withstand handling and implantation and, if necessary, mechanical stimulation. As is the case with other materials, good processing properties and sufficient availability on the market are also desirable.

Collagen has been widely studied in TERM as it is the major natural building material for most tissues and organs and collagen scaffolds thus show very high biomimetic properties. Therefore, collagen and gelatin, the denatured form of collagen, are discussed in more detail below. However, collagen is only available in limited quantities, its processing poses major challenges, and processed collagen exhibits only moderate mechanical properties under physiological conditions. Therefore, many other biocompatible polymers have become the focus of research for scaffold materials. Common organic biomaterials are naturally occurring polymers such as alginate, cellulose, collagen, chitosan, chitin, gelatin, hyaluronic acid, and silk fibroin and synthetic polymers such as polycaprolactone, polyglycolide, polylactic acid, and poly(lactic-co-glycolic acid). Chitosan is a cationic and hydrophilic polysaccharide, which is derived from chitin, which is the second most abundant naturally occurring polysaccharide after cellulose, but is poorly soluble [33]. Alginate and hyaluronic acid are anionic and hydrophilic polysaccharides [34,35]. All three polymers are characterized by their high availability, good processability, and biodegradability. Cellulose is a polysaccharide produced by bacteria and plants, which is characterized by its excellent mechanical properties, but is not biodegradable in the human body [36]. Silk fibroin is a protein fiber spun by Bombyx mori, which has very good mechanical properties and a low degradation rate in the human body [37]. Polycaprolactone is currently the most widely used synthetic polymer in the field of TERM, as it is characterized by its good processing properties and a low degradation rate [38]. Polycaprolactone and other degradable polyesters are hydrophobic when compared to naturally occurring biopolymers and thus exhibit comparatively poor cell adhesion, which may require surface treatment for use in TERM [39]. In addition to the biopolymers listed here, which are widely used in TERM, very specific biopolymers such as smart biomaterials that interact with biological systems are also being studied [40,41]. Important inorganic biomaterials are bioactive glass [42] and hydroxyapatite [43].

### 3.2. Collagen

Collagens were intensively studied for use in medicine in the 20th century [44]. By the end of the 20th century, collagens were displaced by the development of new synthetic polymers such as polylactides, polyglycolic acids, and polylactide-co-glycolides [45]. Since the newly developed polymers have often failed to meet medical expectations such as biocompatibility, biodegradability, and biomimetics, collagen-based materials are gaining renewed importance, especially in the context of TERM.

Currently, 28 collagen types are known in humans. With the exception of cartilage tissue, all tissues contain fibrillar type I collagen, being the most abundant in the human body. Other rarer types perform specialized tasks (e.g., type II in hyaline cartilage) or complement type I collagen in specific parts of the body (e.g., types I and III in skin, types I and VI in connective tissue) [46]. In the following sections, the terms collagen and type I collagen are used synonymously.

The collagen molecule is composed of three polypeptide helices. This right-handed triple helix (tertiary structure) has a diameter of about 1.5 nm and a length of about 300 nm [47]. Collagen molecules are naturally found in the ECM. They self-assemble into larger ordered structures such as microfibrils, fibrils, and fibers (quaternary structures) [48]. This self-assembly or fibrillogenesis is part of natural tissue formation in vivo. Likewise, fibrillogenesis can be initiated in vitro [49,50]. A characteristic feature of assembled collagen molecules is that the ends of adjacent collagen molecules are aligned by a specific offset to each other. This offset is called the D-period and ranges from 60 to 73 nm depending on the tissue type [51].

The helical tertiary structure of collagen molecules is irreversibly destroyed when a specific temperature, the denaturation temperature, is exceeded. Dispersed collagen molecules in water denature at a temperature as low as 37 °C [52] The denaturation temperatures of collagenous quaternary structures are much higher (up to 120 °C) and are strongly dependent on their water content [53]. Besides heat, collagen can also be denatured by mechanical influences [54] or organic solvents [55]. After denaturation, the helix is (partially) destroyed and the three polypetide chains are (partially) disordered. Denatured collagen is known as gelatin. It dissolves in an aqueous environment at a temperature of approximately 40 °C. From an energetic point of view, the polypetide chains would adopt the helical structure of collagen upon subsequent cooling (complete renaturation). However, only partial renaturation occurs, since peptides were entangled and thus cannot completely rearrange to the helical structure [56].

Both collagen and gelatin contain amino acid sequences for cell adhesion. Due to their different tertiary and quaternary structures, these amino acid sequences have different accessibility. These sequences are RGD sequences in gelatin (arginine, glycine, and aspartic acid) and GFOGER sequences in collagen (glycine, phenylalanine, hydroxyproline, glycine, glutamic acid, and arginine) [57]. Studies have shown a positive effect on cellular behavior to both the RGD sequence of gelatin [58,59] and the GFOGER sequence of collagen [60,61]. Other studies indicate that collagen-based scaffolds may have beneficial effects on cellular behavior compared to gelatin-based scaffolds [62,63].

Collagen is considered particularly suitable for TERM as it is the most abundant protein in natural ECM and therefore collagen has a very high biomimetic potential as a biomaterial.

Collagen is derived from animal sources such as rat tails [64], calf skin [65], horse tendons [66], and other animal tissues. Within the last few years, marine animals as a source of collagen have increasingly been studied [67]. Moreover, transgenic plants as a collagen source have been studied [68]. Acid-soluble collagen is extracted from animal tissues using acids. To extract acid-insoluble collagen, the animal tissue is treated with proteolytic enzymes such as pepsin [69]. This treatment produces the collagen derivative atelocollagen, in which the N- and C-terminal non-helical sequences of collagen are removed, enabling solubility for the extraction process. In some publications, atelocollagen is considered to have lower immunogenicity compared to tropocollagen, however, no comparative studies on the immunogenicity of tropocollagen and atelocollagen are available and no reliable statement can be made [70].

Three-dimensional porous collagen scaffolds are therefore state-of-the-art. They are fabricated using electrospinning [71,72], 3D printing [73,74], freeze-drying [75,76], phase separation (in combination with polycaprolactone [77], in combination with hydroxyapatite [78]), and salt leaching [79,80].

Collagen-based membranes such as Chondro-Gide^®^ (Geistlich Pharma AG, Wolhusen, Switzerland), Cartimaix (Matricel GmbH, Herzogenrath, Germany), Novocart^®^ Basic (B. Braun SE, Melsungen, Germany), MaioRegen™ (Finceramica, Faenza, Italy) and collagen-based gels such as ChondroFiller (meidrix biomedicals GmbH, Esslingen am Neckar, Germany) and CaReS^®^-1S (Arthro-Kinetics AG, Tübingen, Germany) are commercially available and approved for the treatment of cartilage defects.

## 4. Isotropy and Anisotropy

Isotropy means uniformity in all directions, whereas anisotropy means dependence on directions. Consequently, an isotropic scaffold shows a similar or identical response regardless of the direction of, for example, an applied force. Anisotropic scaffolds, on the other hand, are characterized by directional properties. Thus, parameters such as porosity or strength depend on the direction.

The idea in using scaffolds for TERM is to mimic the natural ECM. Most tissues in the human body have a collagen fiber based structure, and thus, directional fiber reinforcement, which, among other things, gives them their robust and shock-absorbing properties. Anisotropy is found from the macroscopic to the molecular level, as collagen fibers themselves have anisotropic properties as their molecules are aligned in the direction of the fiber axis. Therefore, ECMs with aligned and parallel collagen fibers (Figure 3) have anisotropic properties, for example, in terms of their mechanical load capacity. These anisotropic properties have to be mimicked by biomimetic scaffolds (e.g., by mimicking the orientation of the collagen fibers of the tissue to be regenerated).

## 5. Isotropic Scaffold Fabrication

The range of properties of the scaffolds developed for TERM is very wide as they are tailor-made for specific applications and tissues by using a variety of biomaterials as well as fabrication methods. Scaffolds may be composed of a graphene single-layer [81], electroactive elastomer actors [82], and metals [83]. For the fabrication of three-dimensional open porous scaffolds, five main conventional fabrication methods are applied: 3D printing, freeze drying, phase separation, salt leaching, and gas foaming (Figure 4).

Phase separation involves the conversion of a polymer solution into a two-phase heterogeneous material system in which polymer-enriched regions and solvent regions are separated. Phase separation can be initiated thermally by cooling the polymer solution [84] or by mixing the polymer solution with a non-solvent or coagulant [85]. After phase separation, the solvent is removed from the two-phase system by freeze-drying. Phase separation is mainly used to prepare polylactide scaffolds, which have pore sizes of 100 µm [86], 100 and 200 µm [87], and 15 to 55 μm [88]. In comparison to other fabrication methods and in the context of cell migration, the mean pore size of these scaffolds is small and influencing the pore geometry is limited. Currently, this method is not widely used to prepare scaffolds from biopolymers such as chitosan, collagen, and alginate. Nevertheless, it should be noted that the mechanism of phase separation is also used in almost all other fabrication methods, since the polymer is often dissolved in a solvent and separated from the solvent in a subsequent step.

Various additive manufacturing methods can be used to prepare scaffolds such as “3D printing” (bonding of particles by binder application), “fused deposition modeling” (solidification of extruded thermoplastic threads by cooling), “stereolithography” (solidification of resins by photopolymerization using lasers), “selective laser sintering/melting” (bonding of particles by melting/sintering using lasers) and “3D plotting” (solidification of extruded hydrogel threads by cooling or coagulating) [89]. The main advantage of all additive manufacturing methods when compared with other conventional fabrication methods is the achievable shape variety and complexity of the printed structures and the almost unlimited adjustability of the pore size. Their main disadvantage, however, is that the specific additive manufacturing method chosen determines the material and its solidification or bonding mechanism, and therefore the biopolymer cannot be selected arbitrarily.

With “3D printing”, all powdery materials can be processed, but an additional binder or adhesive is mandatory to bond the powdery material. With “stereolithography”, typical biomaterials cannot be processed as the method is based on the photopolymerization of liquid monomers or prepolymers. With “fused deposition modeling” and “selective laser sintering/melting” and “fused deposition modeling”, only thermoplastics such as polylactide [90] and polycaprolactone [91] can be processed. Non-meltable biopolymers are almost exclusively processed into scaffolds using “3D plotting”. Similar to “fused deposition modeling”, a polymer solution is extruded and layered to form structures. The “plotted” structures consist, for example, of collagen threads with a diameter of 330 µm and pore size of 260 µm [92], calcium-phosphate-cement threads with a diameter of 200 µm, pore size of 900 µm [93], and chitosan threads with a diameter of 210 µm and pore size of 175 µm [94]. One advantage of “3D plotting” is that substances such as cells [95] and growth factors [93] can be embedded in the threads. The disadvantage of 3D plotting compared to other additive manufacturing methods is that complex structures such as undercuts, overhangs, and bridges can only be printed to a limited extent or not at all, since no temporary support material can be printed.

For the fabrication of scaffolds by means of freeze-drying, a polymer is distributed in a homogeneous solution or heterogeneous suspension/emulsion and poured into a mold. During subsequent cooling, crystals grow in the liquid phase, leading to a system with two separate phases: a crystal phase and a polymer-enriched phase. During subsequent freeze-drying, the crystals are sublimated and a porous polymer framework is formed [96]. The crystals or pores can be influenced in terms of size, shape, and orientation by controlling the liquid, cooling, and drying parameters. For example, this method can be used to prepare collagen scaffolds with pore sizes ranging from 85 to 325 µm [97], alginate scaffolds with pore sizes ranging from 10 to 141 µm [98], and uniaxially oriented gelatin scaffolds with pore sizes ranging from 50 to 500 µm [99].

In chemical gas foaming, a foaming agent is added to a polymer solution and this mixture is poured into a mold. A chemical reaction creates gas bubbles in the cast polymer. After solvent and gas are removed from the polymer, a porous scaffold is present. However, homogeneous pore distribution can only be achieved by stabilizing the gas foaming agent with (toxic) surfactants. Therefore, methods for physical gas foaming have been developed, where the molded and solvent-free polymer is infiltrated with carbon dioxide gas in a pressure chamber. For hydrophilic polymers, an auxiliary solvent is needed to infiltrate the nonpolar carbon dioxide. Upon subsequent depressurization, the gas expands and forms pores within the polymer. The pore geometry can be influenced by controlling the pressure, temperature, and pressure decrease rate [100]. By means of chemical gas foaming, for example, alginate scaffolds with pore sizes ranging from 100 to 400 µm [101], gelatin scaffolds with pore sizes ranging from 280 and 550 µm [102], and calcium phosphate cement scaffolds with pore sizes ranging from 100 to 400 µm [103] can be prepared. Using physical gas foaming, for example, chitosan scaffolds with pore sizes of 30 to 40 µm [104], polycaprolactone scaffolds with pore sizes of 40 to 250 μm [105], and chitosan/gelatin/alginate scaffolds with a pore size of 57 µm [106] can be prepared.

Scaffolds are fabricated by salt leaching by dispersing a porogen (particle) in a polymer solution without dissolving the porogen in the polymer solvent. After the polymer is formed and the solvent is removed, the porogon is dissolved from the polymer, leaving a porous structure. The method limits the choice of polymer because water-soluble inorganic salts are often used as porogen. Hence, aqueous polymer solutions cannot be used as that salt particle would be dissolved too early. Therefore, polymers that dissolve in organic solvents are often applied. The pore size can be adjusted by the particle size of the salts [107]. By salt leaching, for example, polycaprolactone scaffolds with pore sizes from 50 to 500 µm [108], silk scaffolds with pore sizes of about 380 µm [109], and polylactide scaffolds with pore sizes from 166 to 453 µm [110] can be prepared.

In each of the scaffold examples given, one fabrication method was used. In the literature, however, the methods are often combined in order to take advantage of several methods. For example “3D plotting” is combined with electrospinning [111,112], freeze drying with salt leaching [113,114], “fused deposition modeling” with gas foaming [115,116], and thermally induced phase separation with non-solvent induced phase separation [117,118].

Although the mentioned conventional fabrication methods have many specific advantages and are equally characterized by the fact that they can be applied using simple tools, none of the conventional methods can affect the molecular structure of the polymer network of the scaffolds. Only the macroscopic shape of the scaffolds can be influenced, and the polymer network is essentially present as an isotropic structure. However, the goal of biomimetic scaffolds is to mimic the natural ECM and its anisotropic properties as closely as possible. In summary, the characteristic molecular structure or the collagen fiber architecture of the ECM of the individual tissues [119] cannot be prepared by conventional fabrication methods.

## 6. Anisotropic Fiber Scaffold Fabrication

### 6.1. Spinning Methods

Most natural fibers such as cotton fibers [120] and wool fibers [121] as well as most man-made endless fibers (=filaments) such as ultra-high molecular weight polyethylene filaments (Dyneema^®^) [122] and polyester filaments [123] have a molecular structure oriented in the fiber direction, resulting in strongly pronounced anisotropic properties such as a high tensile strength and modulus. If fiber or filament yarns are further processed into scaffolds using textile technologies, the fiber position and orientation can be influenced, and thus a spatially defined molecular structure of the polymer network can be realized.

Spinning methods for the production of continuous filament yarns can be divided into three main types: wet spinning, dry spinning, and melt spinning (Figure 5). In each method, liquid spinning masses are extruded through spinnerets, subsequently drawn and wound on a spool. Subsequently, the dry or consolidated yarn is processed into textile structures by means of textile technologies. The diameter of an individual filament is typically between 10 and 100 µm. In addition to these main types of spinning, electrospinning may be used for the production of nanofiber-based structures. Here, filaments are also produced, but in contrast to the main spinning methods, these filaments have significantly smaller diameters (<1 µm) and are not wound on a spool, but are deposited as a nonwoven on a substrate [124].

In the case of both wet and dry spinning, a polymer is dissolved with a solvent and placed in a spinning tank. In wet spinning, the polymer solution is extruded by pumps through spinnerets into a coagulation bath, in which the polymers of the extruded threads are precipitated by chemical phase separation. Subsequently, the polymer threads are drawn into filaments, washed, dried, and the resulting yarn is wound on a spool. Compared to wet spinning, dry and melt spinning require spinning masses with significantly higher viscosity, since much higher polymer concentrations are used. To homogenize the spinning masses, extruders are employed to transport the polymer solution in the case of dry spinning or the polymer melt in the case of melt spinning to the spinning pumps. By means of pumps, the spinning masses are pushed through the spinnerets and the extruded threads are drawn into filaments. Solidification is based on solvent evaporation (dry spinning) or cooling (melt spinning). Typical polymers for wet spinning are cellulose [125] and polyacynitrile [126] for dry spinning cellulose acetate [127] and for melt spinning polyester [128] and polyamide [129].

For the spinning of some specific polymers, spinning methods need to be adapted, further developed, and combined. Therefore, special spinning methods have been established such as matrix spinning for particularly resistant filaments made of the non-soluble or meltable polytetrafluoroethylene [130], gel spinning for high-strength filaments made of the ultra-high-molecular-weight polyethylene [131], air-gap spinning for improving polymer orientation for filaments made of aramid (e.g., Kevlar^®^) [132], or melt electrospinning for nanofilaments made of non-soluble but meltable polymers [133].

Electrospinning is a commonly used spinning method for direct fabrication of membrane scaffolds or surfaces for cell adhesion, which are made of nano- and micro-fibers. For this purpose, a polymer is dissolved in a solvent and extruded through a cannula. The extruded thread is accelerated in an electric field between the cannula and the substrate, whereby the thread is drawn and the solvent evaporates. The resulting nanofilaments, 10 to 1000 nm in diameter, are layered either randomly or directionally on the stationary or rotating substrate to form a membrane [134]. Alignment of the fibers is achieved by spinning on rotating electrodes [135] or in between the gap of two electrodes [136]. However, it should be noted that the influence of orientation is relatively small (i.e., it can typically only be implemented in one direction; orientations perpendicular to the plane, multi-axial orientations or local orientations are very difficult to implement). Since electrospinning has very low material throughput, modified variants such as multiple-needle electrospinning [137], needleless electrospinning [138], or melt electrospinning exist to increase mass throughput [139]. Another major challenge in electrospinning is to improve the mechanical properties of the membrane structures as they generally have very low strength and low ductility. Increasing the strength is achieved by, for example, reinforcing the nanofilaments with a filler material, increasing the orientation of the nanofilaments, and using post-treatment methods such as a heat treatment, post-drawing, or crosslinking [140]. Pore sizes of electrospun membranes are often too small (<100 µm) to allow for the migration of cells into the interior of the structure. Therefore, electrospinning is also modified in terms of increasing pore sizes [141,142]. Due to the random fiber deposition, the molecular structure of the polymer network cannot be influenced, or in the case of modified electrospinning methods, only to a certain extent.

Most spinning methods can produce various types of bi- and multi-component filaments. The components either remain as a composite or one of the two components serves as a temporary carrier and is removed afterward; this is the case for the popular example of microfiber production (diameter 100 to 5000 nm) from the bicomponent type island-in-sea, in which the microfilaments or -fibers (islands) are embedded in a circular matrix (sea) [143]. Island-in-sea yarn types are also used to produce shape memory yarns by embedding a phase changing or temporary phase as islands in a permanent elastic phase as a matrix [144]. The two components can likewise be arranged in other bicomponent types such as side-by-side [145,146,147], core-sheath [148,149,150], and segmented-pie [151,152,153]. To produce multiple bicomponent filaments (bicomponent multifilament), bicomponent spinning packages are applied in which the components are distributed over all individual core-sheath-nozzles of the spinneret. These packages are composed of several plates, each with specific channels and holes (Figure 6D–F). Due to the circularly symmetrical and three-dimensional guidance of the two components through the spinning package, a similar pressure can be set at each core-sheath-nozzle, thus enabling the production of homogeneous bicomponent multifilament yarns.

In wet spinning, spinning solutions with high viscosity (compared to electrospinning and “3D plotting”) are applied to enable fast coagulation or solidification rates. However, for the production of filaments from spinning solutions with low viscosity and therefore slow solidification rates, standard wet spinning cannot be applied, since the solidifying threads would break during transportation through the coagulation bath. Therefore, microfluidic wet spinning (also called hydrodynamic wet spinning or flow focusing) has been established for the production of filaments from spinning solutions with low viscosity [154]. Therefore, flat (two-dimensional) microfluidic systems (Figure 6A–C) are applied, which were originally developed for lab-on-a-chip applications [155]. These systems consist of stacked polydimethylsiloxane layers on a glass support, whereby the internal layers are engraved with channels. In these systems, for example, two components can be extruded side by side or coaxially (core-sheath) to each other through a bicomponent-spinneret, with a coagulation medium as the sheath component and the spinning solution as the core component. As a result, the spinning solution is temporarily stabilized during coagulation and can be transported through a flow channel without breaking to ensure continuous thread formation. After sufficiently long coagulation within the flow channel, the strength of the thread is high enough and the filament exits the microfluidic system for further drying and winding. Microfluidic wet spinning can be used to produce filaments that are difficult or impossible to produce using standard wet spinning methods such as filaments made from globular proteins [156], colloidal particles [157], nanocellulose [158], collagen [159], and filaments with incorporated cells [160]. In addition, this method is used to produce filaments on a small laboratory scale such as for calcium alginate filaments [161], polychromatic filaments [162], and silk filaments [163], although spinning would also be feasible using standard wet-spinning methods. Unlike standard wet spinning, microfluidic wet spinning is not currently used to produce multifilament yarns because the flat two-dimensional microfluidic systems are not suitable for producing multicomponent multifilament threads.

### 6.2. Chitosan Filament Yarns

Standard spinning methods are applied to produce chitosan filament yarns. Chitosan can be dissolved in aqueous acidic solutions with a pH below 6.7 [164]. The solubility of chitosan is based on a protonation of the amino groups. The extruded chitosan threads are precipitated into continuous filaments within an alkaline coagulation bath (e.g., by sodium hydroxide) for wet spinning [165,166,167] and within an alkaline ammonia gas phase for dry spinning [168] by deprotonating the amino groups of the chitosan. To improve the mechanical characteristics of chitosan yarns, spinning solutions can be prepared using ionic liquids [169] and incorporating additives such as chitin [170] and nanocellulose particles [171]. Additionally, it is possible to convert chitosan filaments into chitin (chitosan is derived from the non-soluble chitin) filaments after spinning using acetylation [172].

Chitosan filament yarns are commercially available, for example, from Saturn Bio Tech Co., Ltd. (Gangwon-do, Korea) [173], ChiPro GmbH (Bremen, Germany) [174], and Hismer Bio-Technology Co., Ltd. (Shandong, China) [174].

### 6.3. Collagen Filament Yarns

To produce collagen filaments, acid soluble collagen is dissolved in aqueous acid solutions with a pH of 2 to 3. The extruded collagen threads are precipitated within neutral coagulation media. Precipitation is based on the fibrillogenesis of collagen, which is initiated by increasing the ionic strength and increasing the pH. Since the spinning solution has a low viscosity and the filament solidification or fibrillogenesis takes several minutes (about 5 to 45 min, depending on the method), standard wet spinning methods cannot be applied for filament formation.

The state-of-the-art of collagen spinning is much less advanced when compared to chitosan spinning. Various technical approaches for collagen filament formation have been described in the literature: for discontinuous fiber formation, extruded collagen threads are deposited within a coagulation bath for several minutes [175,176,177,178,179,180,181,182]. However, these short collagen fibers are not suitable for textile processing. Continuous spinning processes are necessary for collagen filament production. Therefore, technical approaches have been developed in which collagen is extruded into the flowing coagulation medium. For technical implementation, conveyor belts [183], pumps generating a flow within the coagulation bath [184], core-sheath-spinnerets with connected tubes in which the collagen thread and the coagulation medium flow simultaneously [185,186] and microfluidic systems in which the thread and coagulation medium also flow simultaneously [159] are used. Likewise, fibrillogenesis of the spinning solution can be initiated by an electric field only. For the technical implementation of filament formation, the spinning solution is extruded onto a rotating pair of electrodes [187]. With all these methods, only single filaments or monofilaments can be produced. However, only multifilament yarns are suitable for further mechanical textile processing since monofilament yarns cannot withstand the forces of further processing and the mass throughput is also limited.

For the production of collagen multifilament yarns, the monofilament-core-sheath-spinneret has to be duplicated into a multifilament-core-sheath-spinneret (analogous to melt spinning packages with multifilament-core-sheath-spinneret, Figure 6D–F). In such a spinneret, the distribution of the spinning solution and the coagulation medium are designed to ensure a homogenous pressure distribution within the multifil.-core-sheath-spinneret and enable simultaneous thread extrusion (Figure 7). Through a connected coagulation tube, the threads are flowing parallel and without sticking to each other as a protective skin is formed as soon as the collagen solution make contact with the coagulation medium. Within the tube, threads are drawn into filaments as a roller or winder pulls the filaments to the exit of the tube. After exiting the tube, the solidified filament bundle passes washing baths, a drying section, and the yarn is finally wound on a spool (Figure 8). These yarns can then be used for further processing such as the production of scaffolds by means of textile technology processing.

### 6.4. Fiber Based Scaffolds by Textile Technologies

Yarns are further processed into 1D, 2D, or 3D textiles using textile technologies by weaving [188], weft knitting [189], warp knitting [190], braiding [191], flocking [192], and fiber based additive manufacturing methods [193] (Figure 9). Due to the application of these textile technologies, the textile structures consist of a spatially defined fiber structure. This is another reason why textile technologies are applied, for example, as efficient manufacturing technologies for lightweight structures with load-adapted fiber orientation based on the spatially defined molecular structure [124].

In weaving, scaffolds are produced by crossing monofilament or multifilament yarns. For example, woven scaffolds are made of polylactide monofilaments (thickness 2.4 mm, pore size 224 µm, by hand loom) [194], alginate multifilaments (thickness 0.4 mm, pore dimensions 390 μm × 320 μm × 104 μm, by weaving machine) [195], and collagen monofilaments (no thickness and pore size specification, by hand) [196]. In the case of weft and warp knitting, scaffolds are produced by creating multiple and connected yarn loops. Compared to woven fabrics, these structures are more elastic due to their deformable loop structure. For example, knitted scaffolds are made of polylactide monofilaments (pore size 600 to 1100 µm, by knitting machine) [197], silk multifilaments (no pore size specification, by knitting machine) [198], and collagen-coated silk monofilaments (pore size 10 to 310 µm, by knitting machine) [199]. Braiding is used to produce tubular structures by crossing filament yarns analogous to weaving. For example, these tubes consist of polylactide multifilaments (pore size 5 to 25 µm) [200], polylactide multifilaments (pore size 150 to 250 µm) [201], and polylactide-co-glycolide filaments (pore sizes 175 to 233 µm) [202]. In contrast to the aforementioned textile technologies, for fiber-based additive manufacturing and flocking, only cut filaments or short fibers with a length of a few millimeters are used. Fiber-based additive manufacturing is based on a similar mechanism as seen in “3D printing”. However, instead of a powder, short fibers are bonded with an adhesive or binder. For example, fiber-based 3D printing can be used to produce chitosan fiber scaffolds with a thickness of 3 mm and a pore size of 100 µm [203]. Electrostatic flocking enables the fabrication of compressively elastic structures consisting of vertically aligned short fibers bonded onto a substrate. This process can be applied, for example, in the fabrication of polyamide fiber scaffolds with a thickness of 1 mm [204] and chitosan fiber scaffolds with a thickness of 2 mm and a pore size of 65 to 310 µm (Figure 10) [205,206].

## 7. Cell and Tissue Alignment in Anisotropic Scaffolds

Typically, cell biological studies regarding the anisotropy of scaffolds evaluate the orientation of cultured cells after a few days or weeks using electrospun fiber membranes. Yi et al. studied cell orientation of smooth muscle cells after they were cultured for three days on an electrospun anisotropic polylactide nanofiber scaffold [207]. It was shown that the cells exhibited an orientation analogous to the fiber orientation of the scaffold. Xie et al. determined a similar relationship of cell orientation and fiber orientation after stem cells were cultured for seven days on an electrospun anisotropic polylactide nanofiber scaffold [208].

In a few preclinical studies, the anisotropy of the newly formed ECM is shown. Garrison et al. fabricated a multilayered electrospun anisotropic nanofiber scaffold [209]. After fibroblast culture for seven days, it was found that the newly formed ECM had an orientation analogous to the fiber orientation of the scaffold (analysis based on fibronectin orientation). Uiterwijk et al. conducted a study using an electrospun nanofiber scaffold in a large animal model (sheep) to analyze a scaffold for tissue engineering of a heart valve. However, in contrast to the hypothesis, a long-term study of 12 months failed to show any effect of fiber orientation on tissue formation (analysis based on collagen orientation) [210]. Owida et al. prepared a three-layered electrospun polylactide nanofiber scaffold resembling the layered structure of natural articular cartilage of the knee [211]. After culturing chondrocytes in each layer for 14 days, both orientation of the cells and orientation of the newly formed ECM resembled the orientation of the respective layer.

Presently, no studies are available regarding the evaluation of the anisotropy of newly formed tissue on open porous anisotropic fiber based scaffolds prepared from yarns and textile processing.

## 8. Conclusions

Many biomedical treatments within the field of TERM are based on the use of scaffolds. Scaffolds play an important role, but still show a large potential for improvement, especially with respect to their biomimetics. The biomimetics and other important parameters such as pore size of the scaffolds can be influenced by both the biomaterial used and the fabrication method. Therefore, scaffold fabrication methods have been established, which can be used to produce complex and open-pore scaffolds. However, most established fabrication methods (3D printing, freeze drying, phase separation, salt leaching, and gas foaming) cannot generate anisotropic properties in the scaffold, which are essentially required for biomimetics. Textile technologies can be applied to generate anisotropic scaffolds. Some approaches for anisotropic scaffold fabrication have already been described in the literature, however, the application of textile technologies has not gained widespread popularity in the field of TERM. Suitable biopolymer yarns (e.g. collagen and chitosan) and textile technologies (weaving, weft knitting, warp knitting, braiding, flocking, and fiber-based additive manufacturing methods) are readily available for the fabrication of scaffolds. Through appropriate combinations of yarns and textile technology, anisotropic fiber based scaffolds can exhibit properties similar to native tissues. Due to the defined fiber placement during fabrication, fiber based scaffolds can be designed hierarchically and according to the load-case of the final application. Especially in contrast to electro-spun scaffolds, cells can migrate into the fiber based structure due to the larger pore size. Nevertheless, there is a continued need for research on studies regarding the effect of anisotropic fiber based structures on cell behavior and tissue growth. With the appropriate yarns and textile technologies, a wide range of three-dimensional open porous anisotropic fiber based scaffolds may be fabricated for analysis of their effect on anisotropic tissue formation.

## Figures and Tables

**Figure 1 ijms-22-09561-f001:**
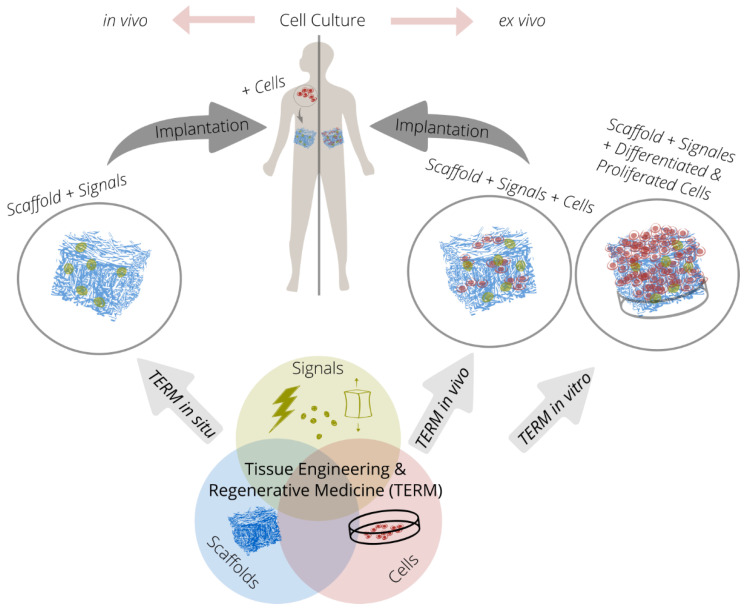
The triad of tissue engineering and regenerative medicine in the context of the in situ, in vivo and in vitro strategy with in vivo and ex vivo cell culture.

**Figure 2 ijms-22-09561-f002:**
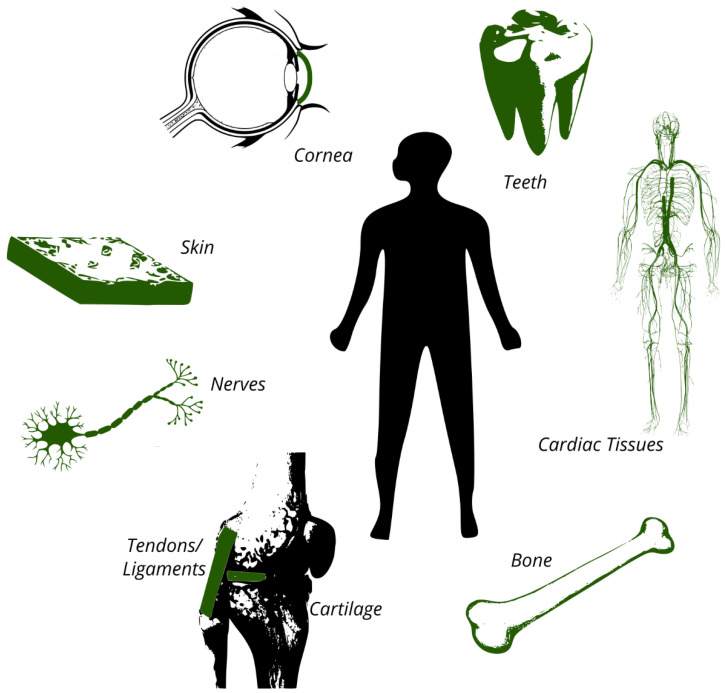
The most important tissues for TERM include cartilage, skin, bone, cornea, nerves, tendons, ligaments, cardiac tissues (arteries, heart valves, and myocardium), and teeth.

**Figure 3 ijms-22-09561-f003:**
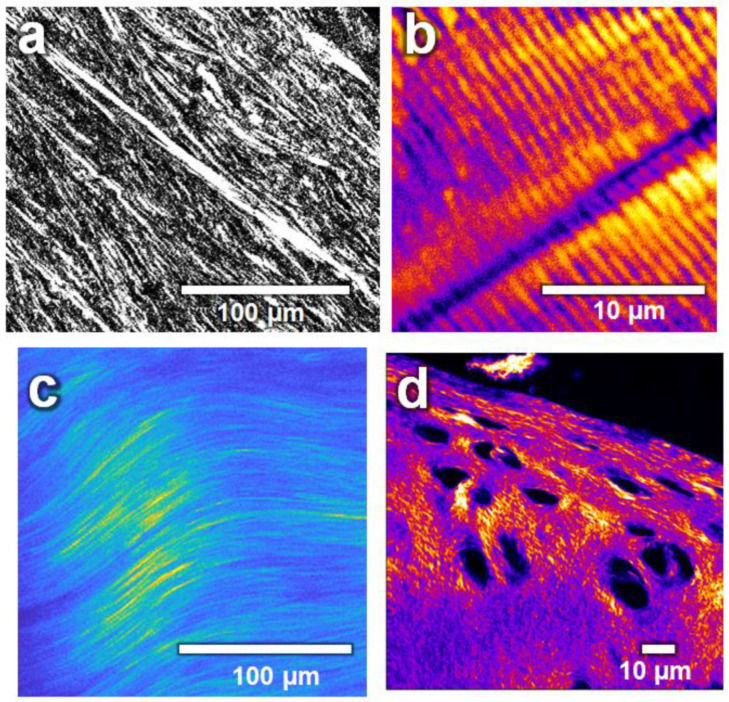
Biological tissues and their collagen fiber orientation imaged by second-harmonic generation microscopy. (**a**) Transverse cut of a human cornea. (**b**) Skeletal muscle from zebrafish (myosin). (**c**) Adult mice-tail tendon. (**d**) Surface cartilage from a knee of a mature horse. Image by BP-Aegirsson CC BY-SA 4.0.

**Figure 4 ijms-22-09561-f004:**
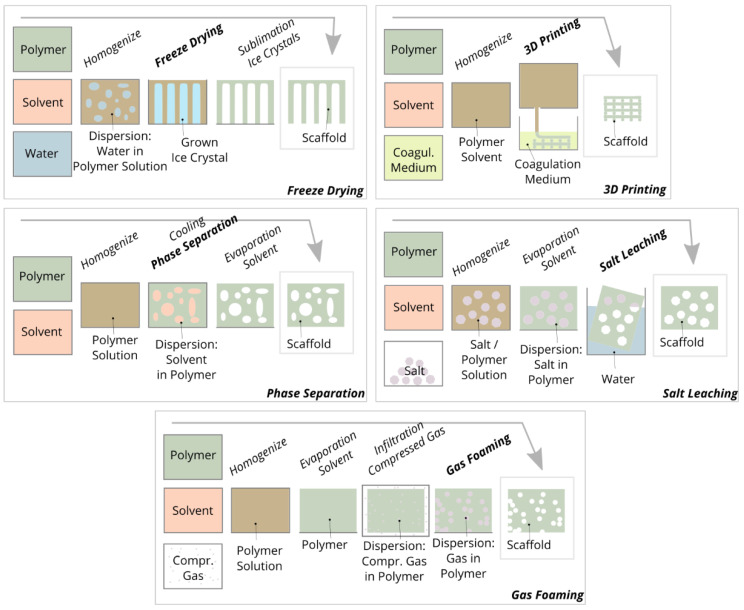
Schematic representation of the five main conventional fabrication methods of scaffolds.

**Figure 5 ijms-22-09561-f005:**
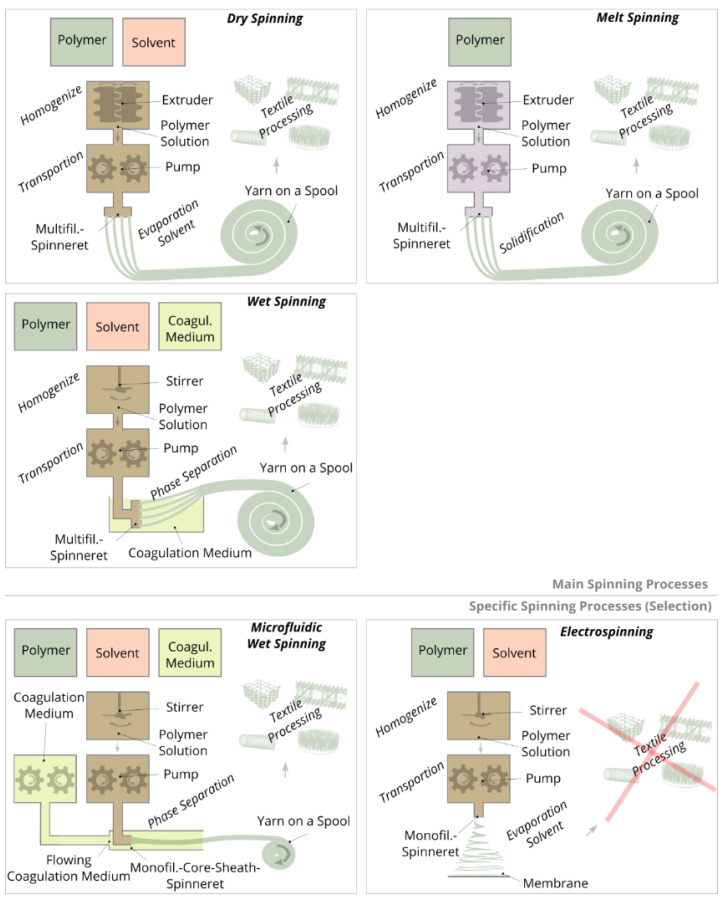
Schematic representation of the three main types of spinning processes: melt, dry and wet spinning for the production of filament yarns; electrospinning and microfluidic wet spinning are additionally shown. In all the processes illustrated, with the exception of electrospinning, yarn spools are produced which can be used for further textile processing. In the electrospinning process membranes or nonwovens are produced.

**Figure 6 ijms-22-09561-f006:**
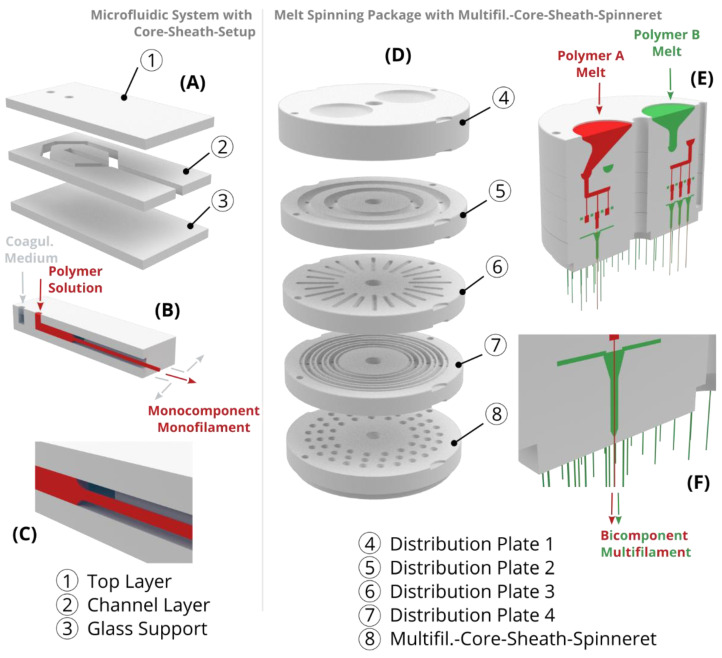
Schematic representation (left) of a microfluidic system with core-sheath-setup consisting of 3 layers, (**A**) shown as exploded view, (**B**) sectional view, and (**C**) detailed view. The inlet and the flow of the coagulation medium (gray-blue) and the polymer solution (red) of the microfluidic system are illustrated. At the exit, the polymer solution is coagulated into a monofilament and the coagulation medium flows off to the side. Schematic representation (right) of a melt spinning package with a multifilament-core-sheath-spinneret, (**D**) shown as exploded view, (**E**) sectional view, and detailed view (**F**). The inlet and the flow of the polymer A (red) and the polymer B (green) of the melt spinning package are illustrated. At the exit, both polymers are coaxially combined to form several bicomponent filaments (multifilament).

**Figure 7 ijms-22-09561-f007:**
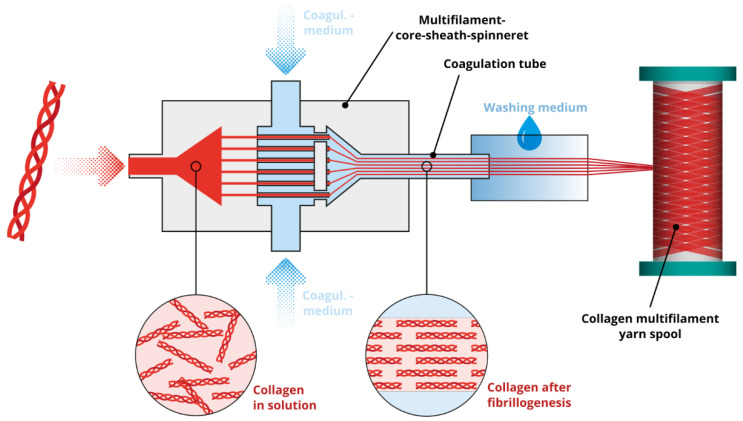
Schematic representation collagen wet spinning with a multifilament-core-sheath-spinneret.

**Figure 8 ijms-22-09561-f008:**
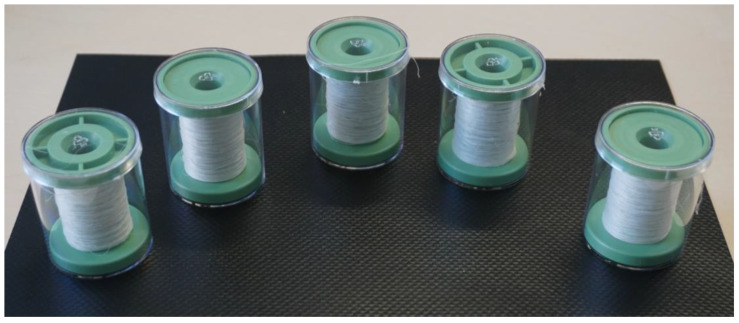
Collagen multifilament yarns wound on spools.

**Figure 9 ijms-22-09561-f009:**
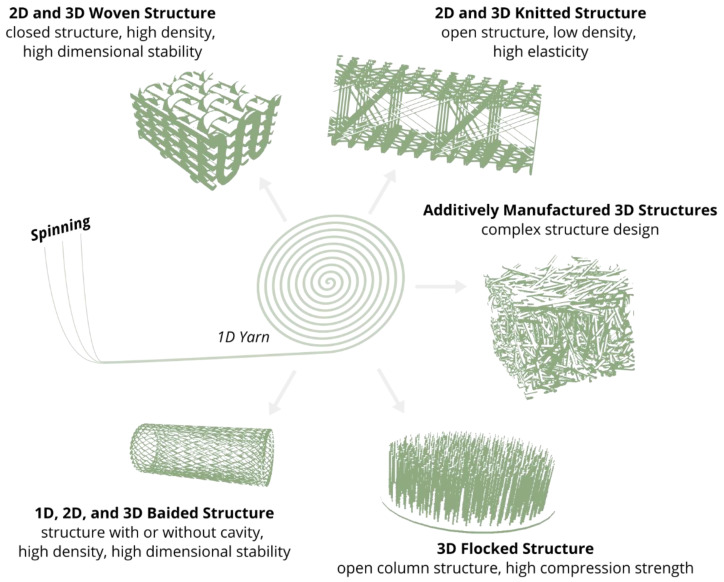
Schematic representation of the main anisotropic fiber scaffold types made by textile technologies.

**Figure 10 ijms-22-09561-f010:**
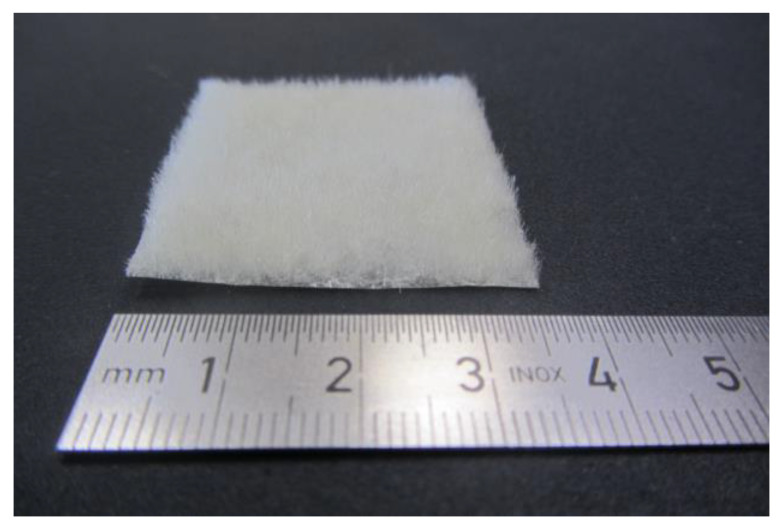
Chitosan fiber scaffold with vertically aligned chitosan fibers [206].

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
