# Peer review of "Isotropic and Anisotropic Scaffolds for Tissue Engineering: Collagen, Conventional, and Textile Fabrication Technologies and Properties"

_ijms, 2021, doi:10.3390/ijms22179561_

Round 1

Reviewer 1 Report

The review article by Tonndorf and co-workers deals with the description of some biomaterials for TERM and the most common scaffold fabrication methods. Although the title focuses on osteochondral defects, the paper is not actually related to this topic (for confirmation of this statement, the reading of the sole conclusion section is enough). Conversely, a broader description of TERM issues and scaffold fabrication approaches is reported. It is strongly suggested to revise the title and to remove the section related to osteochondral defects description (i.e. "TERM of articular cartilage and articular cartilage-bone junction"). Try to refocus the whole review on a broader topic (i.e. edit the title referring to Fabrication approaches of scaffolds for TERM).
Although written in good english and supplied with a well-finished set of explanatory figures, the paper is confusing because the reader expects to find out something that is not reported herein.

Firstly, the Biomaterials paragraph does not take into account the "osteochondral defect" issue. Rather than discussing the most common biomaterials for TERM, those biomaterials specifically meant for articular cartilage treatment should have been described.
In addition, from page 8 on, a list of fabrication methods is reported, without referring to scaffolds specifically made for cartilage repair/regeneration. Furthermore, no explanation of the difference between isotropic and anisotropic materials is provided, while basic consepts like collagen and chitosan structure are reported in excessive detail. 

Minor issue:
Check for typos/English language throughout the manuscript (i.e. lines 47-48, 139, 385, 558).

Reviewer 2 Report

Dear Authors, below are my comments about the submitted manuscript.

  1. The title of the manuscript well conveys with the major concern of the study.
  2. In Figure 2a-b-c and Figure 5a-b-c you shall highlight the statistically significant differences for “at first sight” and comprehension of the data displayed.
  3. Pag. 6, Figure 4 is the same of Figure 3. Please modify.
  4. Pag. 6 lines 176-193: the period should be modified since the database research does not comply with the nowadays standard established by PRISMA [1,2]
  5. Pag. 9 Figure 5: check for accuracy of the caption and the words into the figure.
  6. A major concern in my opinion is the lack of a clear description of the reasons you performed the study. You have to specify the advantages the average reader can catch reading your review.

  1. Liberati, A.; Altman, D.G.; Tetzlaff, J.; Mulrow, C.; Gøtzsche, P.C.; Ioannidis, J.P.A.; Clarke, M.; Devereaux, P.J.; Kleijnen, J.; Moher, D. The PRISMA Statement for Reporting Systematic Reviews and Meta-Analyses of Studies That Evaluate Healthcare Interventions: Explanation and Elaboration. BMJ 2009, 339, doi:10.1136/bmj.b2700.
  2. Moher, D.; Liberati, A.; Tetzlaff, J.; Altman, D.G. Preferred Reporting Items for Systematic Reviews and Meta-Analyses: The PRISMA Statement. International Journal of Surgery 2010, 8, 336–341, doi:10.1016/j.ijsu.2010.02.007.

Reviewer 3 Report

A well written and current manuscript that does not quite match up to the title. The manuscript is an excellent primer for the manufacturing techniques of isotropic and anisotropic scaffolds but appears to arbitrarily reject whole categories of biomaterials without a balanced argument. Despite my criticisms, the article is on the whole well done and would not take much work for me to recommend it for publication.

I feel the title is misleading:

1) A highly detailed description of tissue engineering is discussed at the start of the paper but is not hinted at in the title. This could easily be removed to cut down on length, or at least cut down to a briefer description. (I would recommend cutting down on other "introduction" sections too and getting to the main part of the review sooner.

2) There is very little relating the discussed technologies to the treatment of osteochondral defects. (Having read through the paper I was expecting to see a section describing the advancements in osteochondral defects with these technologies, which does not seem to exist.)

I would suggest changing the title to more accurately represent what the article is actually reviewing.

There also seemed to be some notable omissions in the article.

1) All non-biological biomaterials dismissed out of hand? Could you include a more balanced comparison of both natural vs artificial? There are many exciting papers in this field that do not use natural polymers.

2) Electrospinnign can produce micro and nano fibres (428)

3) There are methods to increase electrospinning deposition (such as multineedle spinning). (434)

4) It is possible to produce aligned fibres with electrospinning using a rotating drum (443)

Other comments:
Figure 4 seems to be missing, it is just repeating the figure above.
Figure 5 has the word "Losungsmittel" instead of solvent in one of the panes.

Round 2

Reviewer 1 Report

The revised manuscript has accomplished the reviewer requests. In this reviewer opinion, just two points need to be further ameliorated to achieve publication in IJMS.

  1. It could be better to shorten the title (now it is effectively focused on the actual topic of the manuscript, but it is very long).
  2. The choice to deepen the description of collagen and chitosan should be motivated. Otherwise the reader may ask "why these two polymers above all the others?"

Reviewer 2 Report

No any other comments

Author Response

The reviewer had no further comments. 

Reviewer 3 Report

I am happy with the way the authors have changed the manuscript contents & title to better suit each other, along with positively addressing all of my comments. I would now recommend this manuscript for publication.

Author Response

The reviewer had no further comments.